# Bridging Graph Position Encodings for Transformers with Weighted Graph-Walking Automata

**Patrick Soga**                                                                 *psoga@nd.edu*
*Department of Computer Science and Engineering*
*University of Notre Dame*

**David Chiang**                                                                 *dchiang@nd.edu*
*Department of Computer Science and Engineering*
*University of Notre Dame*

**Reviewed on OpenReview:** *https://openreview.net/forum?id=tE2NiMGdO7*

## Abstract

A current goal in the graph neural network literature is to enable transformers to operate on graph-structured data, given their success on language and vision tasks. Since the transformer's original sinusoidal position encodings (PEs) are not applicable to graphs, recent work has focused on developing graph PEs, rooted in spectral graph theory or various spatial features of a graph. In this work, we introduce a new graph PE, Graph Automaton PE (GAPE), based on weighted graph-walking automata (a novel extension of graph-walking automata). We compare the performance of GAPE with other PE schemes on both machine translation and graph-structured tasks, and we show that it generalizes and connects with several other PEs. An additional contribution of this study is a theoretical and controlled experimental comparison of many recent PEs in graph transformers, independent of the use of edge features.

## 1 Introduction

Transformers (Vaswani et al., 2017) have seen widespread use and success in recent years in deep learning, operating on numerous types of data. It is natural to also apply transformers to graph-structured data, treating the input graph's vertices as a bag of tokens. There are two important issues to be resolved for a proper formulation of a graph transformer: first, how should a graph's edges and edge features be taken into account in the transformer architecture, and second, what is an appropriate position encoding (PE) for a transformer operating on a graph? We focus on answering the latter by using a graph automaton to compute the PEs for a graph transformer.

Designing PEs for graph transformers is not a new topic in the graph neural network (GNN) literature. Broadly speaking, following the terminology used by Zhang et al. (2019) to describe approaches in developing graph convolutional networks, PEs can be categorized as being (1) spectral or (2) spatial in nature. Spectral methods leverage the graph Laplacian, an highly useful descriptor of key structural features of the graph that are able to describe and model phenomena such as heat diffusion and electrostatic interactions. In contrast, spatial methods use local, node-level features to help the transformer differentiate nodes in the graph. Examples of these features include node degree, shortest-path distance, and self-landing probability during a random walk.

In this work, we propose GAPE (Graph Automaton PE), a PE scheme that is inspired by neither spectral nor spatial methods but rather weighted graph automata. GAPE provides a more principled approach with connections to various previous PEs while performing comparably. We show connections between GAPE and other PE schemes, including the sinusoidal encodings of Vaswani et al. (2017), providing a more satisfying theoretical basis for its use as a generalization of PEs from strings to graphs. Further, although we do not

provide experimental results, GAPE is able to give PEs for directed graphs unlike spectral methods which assume undirected graphs.

In addition, GAPE is able to provide distributed representations for use as PEs. Other graph PE schemes have different strategies for delivering a $k$-dimensional encoding; spectral methods might use the first $k$ eigenvectors of the Laplacian (Dwivedi et al., 2020), while spatial methods might consider random walks of up to $k$ steps (Dwivedi et al., 2022). These are not distributed representations, in the sense that particular dimensions of the PE of a node correspond to particular features of that node in the graph. One consequence of this is that the choice of $k$ may depend on the size of the graph – for example, the graph might not even have $k$ eigenvectors.

To outline our contributions, GAPE (1) can theoretically and empirically simulate sinusoidal PEs and achieves nearly the same BLEU score on machine translation (MT), (2) has mathematical connections with other graph PEs including personalized PageRank (Page et al., 1998) and the Laplacian encoding by Dwivedi et al. (2020) under certain modifications, (3) competes with other recent PEs on several graph and node-level tasks, and (4) gives a $k$-dimensional distributed representation for any desired $k$. In addition, while we do not provide experiments, GAPE is able to encode directed graphs, unlike spectral PE methods. Further, we provide a controlled experimental comparison of recent position encodings independent of the use of edge features.

## 2 Related Work

Spectral methods use the eigenvalues and eigenvectors of a graph's Laplacian matrix to construct a PE. Informally, a graph's Laplacian matrix can be thought to measure the "smoothness" of functions defined on the graph, and its eigenvectors happen to be the smoothest functions. The eigenvectors can then be interpreted as descriptors of how information propagates through the graph. For example, Laplacian eigenvectors can be used for graph clustering (Fiedler, 1973; Shi & Malik, 2000), modeling heat diffusion (Coifman & Lafon, 2006), and solving the max-flow/min-cut problem (Chung, 1997). Given these traits, deriving a PE based on the eigenvalues and eigenvectors of the graph Laplacian is well-motivated.

Dwivedi & Bresson (2021) draw on ideas from Belkin & Niyogi (2003) and use the smallest $k$ eigenvectors of the graph Laplacian according to their associated eigenvalues as PEs. Dwivedi et al. (2020) also claim that these eigenvectors are a generalization of the original Transformer's sinusoidal encodings. We test the effectiveness of these eigenvectors as PEs in an MT context in Section 4. Kreuzer et al. (2021) take a more comprehensive approach, employing an attention-based method to produce a PE informed by potentially the entire spectrum of the graph Laplacian. However, save for augmentations by Zhang et al. (2021), these methods restrict the graph transformer to operating on undirected graphs. This is to ensure that each graph's Laplacian matrix is symmetric and therefore has real eigenvalues.

In contrast, spatial methods utilize local node properties such as node degree (Ying et al., 2021), self-landing probability during a random walk (Dwivedi et al., 2022), and shortest path distance (Li et al., 2020; Ying et al., 2021; Mialon et al., 2021). These methods improve the performance of graph transformers and GNNs in general, and they are less theoretically motivated than the spectral methods, leading to specific cases where their effectiveness may be limited. For example, in regular graphs and cycles, a node's degree is not enough to uniquely identify its position. Other techniques such as those involving random walks are highly local; PEs such as one by Dwivedi et al. (2022) require a choice of fixed neighborhood size to walk. To our knowledge, most PEs such as those currently discussed fall into the spectral or spatial categories with no overall framework connecting them.

Regarding the use of finite automata in neural networks, Johnson et al. (2020) develop a Graph Finite-State Automaton (GFSA) layer to learn new edge types. They treat learning the automaton's states and actions as a reinforcement learning problem and compute a policy governing when the automaton halts, returns to its initial state, or adds an edge to its output adjacency matrix. The GFSA layer can successfully navigate grid-world environments and learn static analyses of Python ASTs. A key difference between the GFSA layer and GAPE is that GAPE is primarily concerned with generating node PEs rather than learning new edge types. Further, the GFSA layer sees its most efficient use on DAGs or graphs with few initial nodes whereas GAPE is intended to be used on arbitrary graphs.

## 3 GAPE: Graph Automaton Position Encoding

### 3.1 Weighted graph-walking automata

In this section, we define a weighted version of graph-walking automata (Kunc & Okhotin, 2013). Graph-walking automata, as originally defined, run on undirected, node-labeled graphs; the graphs also have a distinguished initial node, and for every node, the incident edges are distinguished by labels called "directions." Here, we consider directed graphs (and handle undirected graphs simply by symmetrizing them). Our graphs do not have initial nodes.

**Definition 1.** Let $\Sigma$ be a finite alphabet. A *directed node-labeled graph* over $\Sigma$ is a tuple $G = (V, \ell, A)$, where

- $V = \{1, \ldots, n\}$ is a finite set of nodes;

- $\ell \colon V \to \Sigma$ maps nodes to labels;

- $A \in \mathbb{R}^{n \times n}$ is an adjacency matrix, that is, $A_{ij} = 1$ if there is an edge from $i$ to $j$, and $A_{ij} = 0$ otherwise.

If $\Sigma = \{1, \ldots m\}$, we can also think of $\ell$ as a matrix in $\{0, 1\}^{m \times n}$ such that each column has a single 1.

**Definition 2.** Let $\Sigma = \{1, \ldots, m\}$ be a finite alphabet. A *weighted graph-walking automaton (WGWA)* over $\Sigma$ is a tuple $M = (Q, S, \alpha, \mu, \tau)$, where

- $Q = \{1, \ldots, k\}$ is a finite set of states;

- $S \subseteq Q$ is a set of starting states;

- $\alpha \in \mathbb{R}^{k \times m}$ is a matrix of initial weights;

- $\mu \colon \Sigma \to \mathbb{R}^{k \times k}$ maps labels to matrices of transition weights;

- $\tau \in \mathbb{R}^{k \times m}$ is a matrix of final weights. [1]

If $m = 1$, then for simplicity we just write $\mu$.

**Definition 3.** Let $M$ be a WGWA and $G$ be a directed graph. A *configuration* of $M$ on $G$ is a pair $(q, v)$, where $q \in Q$ and $v \in V$. A *run* of $M$ on $G$ from $(q, u)$ to $(r, v)$ with weight $w$, where $q \in S$, is a sequence of configurations $(q_1, v_1), \ldots, (q_T, v_T)$, where $(q_1, v_1) = (q, u)$, $(q_T, v_T) = (r, v)$, $A_{v_t, v_{t+1}} = 1$ for all $t$, and

$$w = \alpha_{q, \ell(v_1)} \left( \prod_{t=1}^{T-1} \mu_{q_t, q_{t+1}} \right) \tau_{r, \ell(v)}. \tag{1}$$

We can think of a graph automaton as simulating random walks with state. At each time step, the automaton is positioned at some node and in some state. At time $t = 1$, the automaton starts at some node in state $q$ with weight $\alpha_{q, \ell(v_1)}$. Then at each time step, if the automaton is at node $u$ in state $q$, it either moves to node $v$ and transitions to state $r$ with weight $\mu(\ell(u))_{q,r}$, or halts with weight $\tau_{q, \ell(u)}$.

### 3.2 Position encodings

To encode the position of node $v$ using a WGWA $M$, we use the total weight of all runs of $M$ starting in any configuration and ending in configuration $(r, v)$. Let $\mathcal{R}_{r,v}$ be the set of all runs of $M$ on $G$ ending in configuration $(r, v)$. Define the *forward weights* $P \in \mathbb{R}^{k \times n}$ as

$$P_{r,v} = \sum_{(q_1, v_1), \ldots, (q_T, v_T) \in \mathcal{R}_{r,v}} (\alpha \ell)_{q_1, v_1} \, \mu_{q_1, q_2} \cdots \mu_{q_{T-1}, q_T}$$

$$= \sum_{(q_1, v_1), \ldots, (q_T, v_T) \in \mathcal{R}_{r,v}} (\alpha \ell)_{q_1, v_1} \prod_{t=1}^{T-1} \mu_{q_t, q_{t+1}}$$

---

[1] The initial and final weights are commonly called $\lambda$ and $\rho$ (for "left" and "right"), respectively, but since these letters are commonly used for eigenvalues and spectral radii, respectively, we use $\alpha$ and $\tau$ instead.

which is the total weight of all run *prefixes* ending in configuration $(r, v)$. We can rewrite this as

$$P_{r,v} = (\alpha\ell)_{r,v} + \sum_{\substack{(q_1,v_1),\ldots,(q_T,v_T)\in\mathcal{R}_{r,v},\\ T>1}} (\alpha\ell)_{q_1,v_1} \prod_{t=1}^{T-1} \mu_{q_t,q_{t+1}}$$

$$= (\alpha\ell)_{r,v} + \sum_{q,u} \left( \sum_{(q_1,v_1),\ldots,(q_{T-1},v_{T-1})\in\mathcal{R}_{q,u}} (\alpha\ell)_{q_1,v_1} \prod_{t=1}^{T-2} \mu_{q_t,q_{t+1}} \right) \mu_{q,r} A_{u,v}$$

$$= (\alpha\ell)_{r,v} + \sum_{q,u} P_{q,u} \mu_{q,r} A_{u,v}.$$

The matrix version of the above is

$$P = \mu^\top PA + \alpha\ell. \tag{2}$$

Setting $\tau = \mathbf{1}$, then $\text{GAPE}_M(v) = P_{:,v} \circ \tau\ell$ (where $\circ$ is elementwise multiplication) is our PE for $v$. Notice that whether a particular node is able to be assigned a PE does not depend on $k$, unlike the random walk encoding (Dwivedi et al., 2022) or Laplacian eigenvector PE (Dwivedi et al., 2020). The columns of $P$ can be viewed as a distributed representation in the number of states of the WGWA. It should also be noted that one could in principle replace $A$ with any weighted adjacency matrix, which we do in Section 3.3 when connecting GAPE with other PEs. Doing so results in a minor conceptual change to GAPE, scaling the transition weights of the WGWA for each of its runs.

To solve Eq. (2) for $P$, one may use the "vec trick" (Petersen & Pedersen, 2012, p. 59–60):

$$\text{vec}\,P = \text{vec}\,\mu^\top PA + \alpha\ell$$
$$= (A^\top \otimes \mu^\top)\,\text{vec}\,P + \alpha\ell$$
$$(I - A^\top \otimes \mu^\top)\,\text{vec}\,P = \alpha\ell$$

where vec flattens matrices into vectors in column-major order, and $\otimes$ is the Kronecker product. However, solving the above linear system directly for $\text{vec}\,P$ runs in $O(k^3 n^3)$ time due to inverting $(I - A^\top \otimes \mu^\top)$, which is impractical for even small graphs given a large enough batch size. We need a more efficient method.

Note that solving for $P$ in Eq. (2) is essentially computing the graph's stationary state, or fixed point, according to the weights of runs taken by the WGWA. A similar mathematical approach is taken by Park et al. (2022) and Scarselli et al. (2009) who compute fixed points of graphs outside of the context of automata. Unlike these approaches which attempt to approximate the fixed point for tractability (Park et al., 2022) or because the fixed point may not be guaranteed (Scarselli et al., 2009), we compute the fixed point exactly using the Bartels-Stewart algorithm (Bartels & Stewart, 1972): an algorithm for solving Sylvester equations (Horn & Johnson, 2013, p. 111), a family of matrix equations to which Eq. (2) belongs.

Using this algorithm, Eq. (2) can be solved more efficiently than using matrix inversions in $O(n^3 + k^3)$ time. To solve Eq. (2), we use an implementation of the Bartels-Stewart algorithm from SciPy (Virtanen et al., 2020). This solver is unfortunately non-differentiable, and so, unless otherwise indicated, we randomly initialize $\mu$ and $\alpha$ using orthogonal initialization (Saxe et al., 2014) without learning them.

In practice, if $x_v \in \mathbb{R}^d$ is the feature vector for $v$ where $d$ is the number of node features, then we use $x'_v = x_v + \text{GAPE}_M(v)$ as input to the transformer. If $k \neq d$, then we pass $\text{GAPE}_M(v)$ through a linear layer first to reshape it. Further, because, in Eq. (2), $P$ is not guaranteed to converge unless $\rho < 1$ where $\rho$ is the spectral radius of $\mu$, we scale $\mu$ by an experimentally chosen "damping" factor $\gamma < 1$.

### 3.3 Connection with other PEs

We move on draw connections between GAPE and other PEs.

### 3.3.1 Sinusoidal encodings

For any string $w = w_1 \cdots w_n$, we define the *graph* of $w$ to be the graph with nodes $\{1, \ldots, n\}$, labeling $\ell(1) = 1$ and $\ell(i) = 2$ for $i > 1$, and an edge from $i$ to $(i+1)$ for all $i = 1, \ldots, n-1$.

**Proposition 1.** *There exists a WGWA $M$ such that for any string $w$, the encodings $GAPE_M(i)$ for all nodes $i$ in the graph of $w$ are equal to the sinusoidal PEs of Vaswani et al. (2017).*

*Proof.* If $G$ is the graph of a string, the behavior of $M$ on $G$ is similar to a unary weighted finite automaton (that is, a weighted finite automaton with a singleton input alphabet), which DeBenedetto & Chiang (2020) have shown can recover the original sinusoidal PEs. Let

$$\alpha = \begin{bmatrix} 0 & 0 \\ 1 & 0 \\ 0 & 0 \\ 1 & 0 \\ \vdots & \vdots \end{bmatrix} \qquad \mu = \begin{bmatrix} \cos\theta_1 & \sin\theta_1 & 0 & 0 & \cdots \\ -\sin\theta_1 & \cos\theta_1 & 0 & 0 & \cdots \\ 0 & 0 & \cos\theta_2 & \sin\theta_2 & \cdots \\ 0 & 0 & -\sin\theta_2 & \cos\theta_2 & \cdots \\ \vdots & \vdots & \vdots & \vdots & \ddots \end{bmatrix} \qquad \tau = \begin{bmatrix} 1 & 1 \\ 1 & 1 \\ 1 & 1 \\ 1 & 1 \\ \vdots & \vdots \end{bmatrix}$$

where $\theta_j = -10000^{-2(j-1)/k}$. Then the PE for node $i$ is $(\alpha_{:,1})^\top \mu^i$, which can easily be checked to be equal to the original sinusoidal PEs. $\square$

To verify the above proposition, we ran an MT experiment benchmarking several graph PE schemes and compared their performance with GAPE using the open-source Transformer implementation Witwicky,[2] with default settings. Results and a complete experimental description are below in Section 4.

### 3.3.2 Laplacian eigenvector encodings

Next, we turn to connect GAPE and LAPE (Dwivedi et al., 2020), which is defined as follows.

**Definition 4.** Define the graph Laplacian to be $L = D - A$ where $D_{vv}$ is the degree of node $v$. Let $V$ be the matrix whose columns are some permutation of the eigenvectors of $L$, that is,

$$LV = V\Lambda$$

where $\Lambda$ is the diagonal matrix of eigenvalues. Then if $k \leq n$, define

$$\text{LAPE}(v) = V_{v,1:k}.$$

**Remark 2.** If we assume an undirected graph and use the Laplacian matrix $L$ in place of $A$, then there is a WGWA $M$ with $n$ states that computes LAPE encodings. Namely, let $m = 1$, $\alpha = \mathbf{0}$, let $\mu$ be constrained to be diagonal, and let $\tau_{q,1} = 1$ for all $q$. Modifying Eq. (2), we are left with solving $P = \mu P L$ for $P$.

But observe that $\mu = \Lambda^{-1}$ and $P = V^\top$ are a solution to this equation, in which case $\text{GAPE}_M(v) = \text{LAPE}(v)$.

While the above is true, some caveats are in order. First, the choice of $\mu$ does depend on $L$ and therefore the input graph. Second, the theoretical equivalence of GAPE and LAPE requires using the graph Laplacian in place of $A$, which is less interpretable in an automaton setting. Third, since we require $\alpha = \mathbf{0}$, then the system $(I - A^\top \otimes \mu) \text{vec} P = \alpha \ell$ is no longer guaranteed a unique solution, and so there are also trivial solutions to $P = \mu P L$ that need to be removed from consideration when solving for $P$. The connection between GAPE and LAPE is therefore largely formal.

### 3.3.3 Personalized PageRank & random-walk encoding

Next, we discuss a connection between GAPE and Personalized PageRank (PPR) and the random-walk PE (RW) used by Dwivedi et al. (2022), which is a simplification of work by Li et al. (2020).

As described by Zhang et al. (2016), the $v$th entry of the PPR vector $\text{PPR}(u)$ of a node $u$ is the probability that a random walk from $u$ reaches $v$ with some damping value $\beta \in [0, 1]$. That is, at each time step, the random walker either restarts with probability $\beta$ or continues to a random neighbor with probability $1 - \beta$.

---

[2]https://github.com/tnq177/witwicky

Table 1: RW vs PPRP with min-max normalization to bring both PEs to similar scales. Dataset descriptions and experimental details regading the transformer and its hyperparameters are in Section 4.

| PE scheme | CYCLES ($\uparrow$) | ZINC ($\downarrow$) |
|---|---|---|
| RW | 99.95 | 0.207 |
| PPR | **100.00** | **0.198** |

$\text{PPR}(u)$ can therefore be interpreted as a PE for $u$ where $\text{PPR}(u)_v$ measures the relative importance of $v$ with respect to $u$. Formally, we define

$$\text{PPR}(u) = \beta e_u + (1 - \beta)\pi_u W$$

where $e_u$ is a one-hot vector for $u$ and $W = AD^{-1}$ where $D$ is the degree matrix of the graph. In matrix form, the PPR matrix $\Pi$ is

$$\Pi = \beta I + (1 - \beta)\Pi W. \tag{3}$$

Written this way, we see a similarity between Eq. (3) and Eq. (2).

**Remark 3.** If we replace $A$ with $W$, there is a WGWA $M$ with $n$ states that computes $\Pi$. Namely, replace $A$ with $W$ and set $k = m = n$, $\mu = (1 - \beta)I$, and $\alpha\ell = \beta I$. Then $\text{GAPE}_M(u) = \text{PPR}(u)$ for all nodes $u$.

The choice to replace $A$ with $W$ results in dividing the automaton's transition weights during each run according to each traversed node's out-degree. Under this modification and the substitutions in the above remark, GAPE nearly generalizes PPR, minorly differing in how transition weights are computed.

Next, we draw a connection between GAPE and RW bridged by PPR.

**Definition 5.** Let $u$ be a node in the graph. Then $\text{RW}(u)$ is defined by

$$\text{RW}(u) = \left[W_{uu}, (W^2)_{uu}, \ldots, (W^k)_{uu}\right] \in \mathbb{R}^k.$$

for some experimentally chosen $k$.

In other words, $\text{RW}(u)_i$ is the probability that a random walk of length $i$ starting at $u$ returns to $u$. Now, Eq. (3) can be seen as a way of calculating something analogous to RW. Since $\Pi_{u,v}$ is the probability of a node $u$ landing on node $v$ during an infinitely long random walk, then $\Pi_{u,u}$ is the self-landing probability of a random walker starting at node $u$, which captures similar information as $\text{RW}(u)$. The only difference is that $\text{RW}(u)$ is determined by random walks of a fixed length. It turns out we can recover very similar encodings as $\text{RW}(u)$ by taking the diagonals of the PPR matrix for successive powers of $W$. That is, if $\Pi^{(i)}$ is the PPR matrix using $W^i$, then we can define the PPR Power (PPRP) encoding of a node $u$ by

$$\text{PPRP}(u) = \left[\Pi^{(1)}_{uu}, (\Pi^{(2)})_{uu}, \ldots, (\Pi^{(k)})_{uu}\right] \in \mathbb{R}^k.$$

Intuitively, $\text{PPRP}(u)_i$ is the self-landing probability of a random walker during an infinitely long random walk with a step size of $i$. Interestingly, while PPRP and RW are not equal, their relative values are nearly identical, and they give nearly identical performance on certain graph tasks as depicted in Fig. 1 and Table 1, respectively. With this, the same postprocessing steps to construct RW can be applied to GAPE through GAPE's connection with PPR to construct PPRP, a PE that is analogous and empirically equivalent to RW.

## 4 Experiments

In this section, we compare GAPE experimentally to the following graph PEs:

**RW** Random Walk (Dwivedi et al., 2022; Li et al., 2020)

**LAPE** LAplacian PE (Dwivedi et al., 2020; Dwivedi & Bresson, 2021)

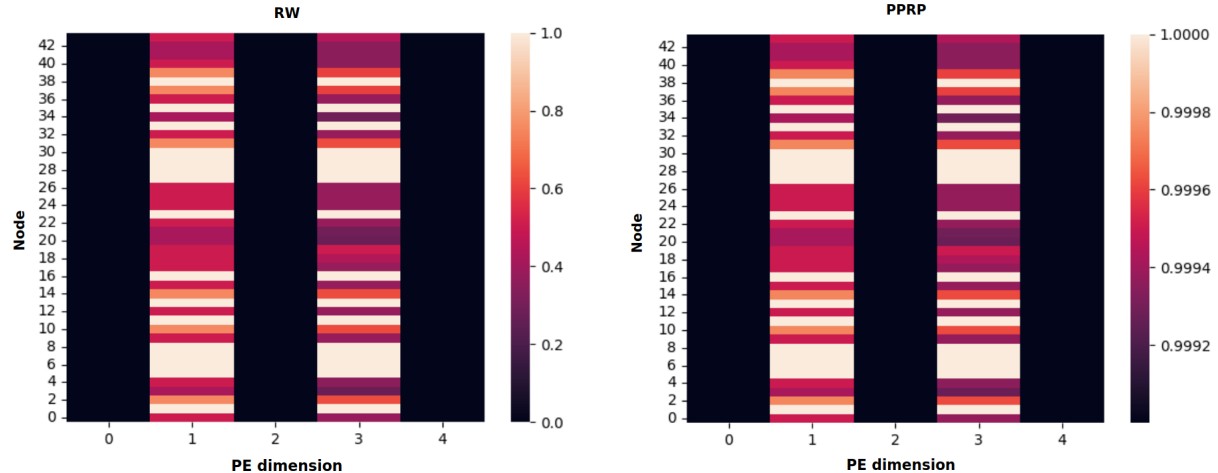

Figure 1: Comparison of RW with PPRP where $k = 5$ and $\beta = 0.999$ for a graph in CYCLES. While PPRP is not calculating self-landing probabilities, the relative differences are the same.

**SA** Spectral Attention (Kreuzer et al., 2021)

**SPD+C** Shortest Path Distance + Centrality (Ying et al., 2021)

Ying et al. (2021) use "Centrality" to refer to node degree. We compare all of the graph PEs with GAPE in two kinds of tasks: MT and graph and node-level tasks.

### 4.1  Machine translation

Our first experiment tests how well these various non-sinusoidal graph PEs perform on MT from English to Vietnamese. To test GAPE, we use the adjacency matrix of the directed path graph of $n$ vertices, where $n$ is the number of tokens in the input sequence, and use $k = 512$. We also initialize $\alpha_{:,1}$ with a normal initialization and set all other columns of $\alpha$ to 0 in order to force the WGWA to start walking the path graph from its first node.

For LAPE, we experiment with two different graphs: the path graph of $n$ vertices and the cycle graph of $n$ vertices. We use the path graph to explore the suggestion that the Laplacian eigenvectors of the path graph are a natural generalization of the sinusoidal encodings (Dwivedi et al., 2020). We also experiment with the cycle graph because its Laplacian eigenvectors form the columns of the DFT matrix (Davis, 1979) which Bronstein et al. (2021) note to share a similar structure with Transformer's original sinusoidal encodings. LAPE with the path graph is denoted by LAPE-P while LAPE with the cycle graph is denoted by LAPE-C. For SA, we use the cycle graph.

However, the sinusoidal encoding for a node $u$ in a sequence does not depend on the length of the sequence, but LAPE for a node $u$ does. So, to try to obtain a better fit to sinusoidal encodings, we try an additional variant of LAPE, which we call LAPE-C$_{10K}$, where the cycle graph is fixed with $10,000$ vertices in which the path graph representing the input sentence can be embedded.

For RW, we use the path graph of 1024 vertices (the transformer's maximum input length) with a 20-hop neighborhood. SPD+C uses the same path graph as RW and uses a shortest path distance embedding size of 256 and a degree embedding size of 2 since any node in the path graph can have a maximum degree of 2.

The results are shown in the first column of Table 2. We see that GAPE achieves a higher BLEU score than every other graph PE and very nearly matches the original sinusoidal PEs of Vaswani et al. (2017), giving empirical evidence confirming Proposition 1 and establishing GAPE as a generalization of sinusoidal PEs. Further, despite previous speculations on the connection between eigenvectors of the cycle graph Laplacian and the sinusoidal encodings, LAPE-P, LAPE-C, LAPE-C$_{10K}$, and SA underperform. While the

eigenvectors of the path and cycle graph Laplacians are indeed sinusoidal, it appears they are unable to recover the performance of the original transformer's PE in practice.

Surprisingly, SPD+C performs poorly despite its shortest path distance encoding being closely related to the relative PE by Shaw et al. (2018); Huang et al. (2019). However, it makes sense that the degree encoding is of little use since there are only two degree types in the path graph: one to denote the beginning and end of the sentence, and another to represent the positions of all other tokens. Under the degree encoding, no two tokens' positions between the start and end tokens are distinguishable.

## 4.2 Graph- and node-level tasks

### 4.2.1 Datasets

We additionally compare performance of GAPE with the other PEs on 7 undirected graph tasks. Below, we give descriptions of each dataset, their importance, and experimental details.

**ZINC** (Irwin et al., 2012) is graph regression dataset with the task of predicting the solubility of various molecules. Given the wide applications of GNNs on molecular data, we thought to use ZINC as it is one of the most popular molecular graph datasets used in the literature. For consistency with the literature on transformer PEs, we use the 12K subset of the full 500K dataset used by Dwivedi et al. (2020) and Kreuzer et al. (2021) and use a batch size of 128.

**CSL** (Murphy et al., 2019) is a graph classification dataset of circular skip link (CSL) graphs. Each CSL graph is 4-regular with $n$ nodes in a cycle and skip links of length $k$ which define isomorphism classes to distinguish. We use the same dataset as Murphy et al. (2019) and follow Chen et al. (2019) choosing $n = 41$ and $k = 10$. Murphy et al. (2019) and Xu et al. (2019) show that the 1-dimensional Weisfeiler-Lehman isomorphism test (1-WL test) (Weisfeiler & Leman, 1968) is unable to distinguish between isomorphism classes of CSL graphs and that message-passing GNNs are at most as powerful as the 1-WL test. So, a PE that succeeds on this dataset can be seen as enhancing the expressive power of its host GNN. There are 150 graphs total, and following Murphy et al. (2019), we perform a 5-fold cross validation split with 5 sets of train, validation, and test data and report the average accuracy. We use a batch size of 5.

**CYCLES** (Murphy et al., 2019; Loukas, 2020) is a cycle detection dataset also meant to test the theoretical power of a given GNN. Loukas (2020) showed the minimum necessary width and depth of a message-passing GNN to detect 6-cycles, and in this task, we are interested in the additive power of each PE in detecting 6-cycles. We use the same train, test, and validation splits as Loukas (2020), using 200 graphs for training, 1000 graphs for validation and 10,000 graphs for test with a batch size of 25.

**CYCLES-V** is also a cycle detection dataset, except with varying cycle lengths. Each graph is a copied from CYCLES, and nodes are added to lengthen cycles and maintain similar graph diameters. Whereas positive samples of CYCLES only contains 6-cycles, positive samples of CYCLES-V contain cycles of length ranging from 6 to 15. We made this adjustment to CYCLES because we wanted to test the generalizability of RW seeing as the choice of random-walk neighborhood size crucially affects its performance; clearly, RW will be able to detect cycles of length $k$ if its neighborhood size is at least $k$. We use the same train, validation, and test splits and batch size as CYCLES.

**PATTERN and CLUSTER** (Dwivedi et al., 2020; Abbe, 2018) are node classification datasets generated using the Stochastic Block Model (Abbe, 2018), which is used to model communities within social networks according to certain hyperparameters such as community membership probability. PATTERN has 2 node classes task while CLUSTER has 7. We use the same splits as Dwivedi et al. (2020) with 10,000 graphs for train and 2,000 graphs for validation and test. We use a batch size of 26 and 32 for PATTERN and CLUSTER, respectively.

**PLANAR** is a new dataset we generate to test the abilities of each PE to help the transformer correctly classify whether a given graph has a planar embedding. We introduce this task as another test of the theoretical power of each PE. For many graphs, non-planarity can be detected by merely counting nodes since a graph is not planar if $|E| > 3|V| - 6$ by Euler's formula. However, exactly determining whether a graph is non-planar requires checking whether there are subgraphs homeomorphic to the complete graph $K_5$

or the utility graph $K_{3,3}$ (Kuratowski, 1930; Wagner, 1937). We use 7,000 graphs for training and 1,500 for validation and testing. On average, each graph has 33 nodes. We use a batch size of 32.

**PCQM4Mv2** is a dataset from the Open Graph Benchmark Large-Scale Challenge (OGB-LSC) (Hu et al., 2021), a series of datsets intended to test the real-world performance of graph ML models on large-scale graph tasks. PCQM4Mv2 is a graph regression dataset with the task of predicting the HOMO-LUMO energy gap of molecules, an important quantum property. As labels for the test set are private, we report performance on the validation set. The train set is comprised of 3,378,606 molecules while the validation set has 73,545 molecules. We use a batch size of 1024. Due to time and hardware constraints, we report metrics based on the 80th epoch during training.

For GAPE, on MT, we used PE dimension $k = 512$ and a damping factor $\gamma = 1$. On the graph-level tasks, we use $k = 32$ and $\gamma = 0.02$. For RW, we choose $k = 20$ on all tasks except for CYCLES-V, where we chose $k = 9$ in order to test its generalizability to longer cycle lengths. For LAPE, we use $k = 20$ on all tasks, following Dwivedi & Bresson (2021). For SA, we use $k = 10$ on all tasks, following Kreuzer et al. (2021).

### 4.2.2 Experimental setup

We use the graph transformer from Dwivedi & Bresson (2021) for our backbone transformer, as it is a faithful implementation of the original Transformer model capable of working on graphs. All models for the graph tasks were subject to a parameter budget of around 500,000 as strictly as possible similar to Dwivedi & Bresson (2021) and Kreuzer et al. (2021). Since there is little consensus on how to incorporate edges and edge features into a graph transformer, we omit the use of edge features in all of our tests for a fair comparison, only using node features.

Across all tasks, we use the Adam (Kingma & Ba, 2015) optimizer. For nearly all tasks, we use 10 layers in the graph transformer, 80 node feature dimensions, 8 attention heads, learning rate of 0.005, reduce factor of 0.5, and patience of 10. For ZINC, we follow Dwivedi et al. (2020) and use a learning rate of 0.007 and patience of 15. CSL is such a small dataset that we opted to shrink the number of transformer layers down to 6 and maintain a parameter budget of around 300,000 for greater training speed and to avoid overparametrization.

SA uses a transformer encoder to compute its PE. For this transformer, we use 1 attention layer, 4 attention heads, and 8 feature dimensions in order to respect the 500,000 parameter budget. For CSL, we reduced the number of attention heads to 1 to respect the 300,000 parameter budget.

For SPD+C, we vary the size of the node degree and shortest-path embeddings according to each dataset since each dataset contains graphs of varying sizes. Degree embedding sizes ranged from 64 to 256, and the shortest-path embedding sizes ranged from 64 to 512.

Table 2 shows performance on graph datasets using neighborhood-level attention. **Bold** numbers indicate the best score of that column up to statistical significance (t-test, $p=0.05$). For MT, best metrics are indicated considering only the non-sinusoidal encodings.

For the graph tasks, Baseline refers to the absence of any PE. For MT, Baseline refers to the original sinusoidal encodings by Vaswani et al. (2017). For each dataset, we take the average of 4 runs each conducted with a different random seed. We report metrics from each dataset's test set based on the highest achieved metric on the corresponding validation set.

### 4.2.3 GAPE variants

We also try several variations of GAPE, with the following normalization and initial weight vector selection strategies.

**GAPE\*** Row-wise softmax on $\mu$ without damping factor $\gamma$.

**GAPE\*\*** Same as GAPE\* but with a column-wise softmax on $\alpha$.

**GAPE$^*_{20}$, GAPE$^{**}_{20}$** Like GAPE\* and GAPE\*\*, respectively, but use $m = 20$ node labels and initialize each initial weight vector $\alpha_{:,\ell}$ to a different random vector for each label $\ell$.

Table 2: Results on MT and Graph Tasks. MT is measured by BLEU score, ZINC and PCQM4Mv2 are measured by mean absolute error (MAE), and the rest are measured by accuracy. OOM stands for Out Of Memory.

| PE scheme | MT (↑) | ZINC (↓) | CSL (↑) | CYCLES (↑) | PATTERN (↑) | CLUSTER (↑) | PLANAR (↑) | CYCLES-V (↑) | PCQM4Mv2 (↓) |
|---|---|---|---|---|---|---|---|---|---|
| Baseline | 32.6 | 0.357 | 0.10 | 50.00 | 83.91 | 71.77 | 50.00 | 73.31 | 0.123 |
| LAPE-P | 17.3 | - | - | - | - | - | - | - | - |
| LAPE-C | 17.4 | - | - | - | - | - | - | - | - |
| LAPE-C$_{10K}$ | 16.4 | - | - | - | - | - | - | - | - |
| LAPE | - | 0.311 | **100.00** | 97.08 | 85.05 | 72.14 | 96.41 | 88.53 | 0.120 |
| SA | 16.9 | 0.248 | 87.67 | 89.52 | 83.61 | 72.84 | 97.45 | 81.11 | 0.158 |
| RW | 20.8 | **0.207** | 93.50 | **99.95** | **86.07** | 72.32 | **98.50** | 90.97 | **0.116** |
| SPD+C | 0.0 | 0.263 | 10.00 | 88.44 | 83.09 | 70.57 | 96.00 | 83.13 | OOM |
| GAPE | **32.5** | 0.251 | 10.00 | 79.03 | **85.99** | 73.40 | 95.97 | 83.86 | 0.122 |
| GAPE* | - | 2.863 | 10.00 | 81.39 | 84.82 | **74.03** | 64.35 | 90.68 | 0.313 |
| GAPE*$_{20}$ | - | 1.339 | 48.67 | 84.66 | 69.95 | 70.57 | 75.12 | 89.01 | 0.556 |
| GAPE** | - | 2.067 | 10.00 | 83.42 | 83.46 | 72.39 | 63.38 | 91.76 | 0.914 |
| GAPE**$_{20}$ | - | 0.837 | 58.00 | 85.43 | 81.53 | 72.06 | 70.48 | **95.12** | 0.440 |
| GAPE**$_{max}$ | - | 0.539 | **100.00** | 86.01 | 83.53 | 72.40 | 71.87 | 95.05 | 0.319 |

Table 3: Runtime for computing PEs ($k = 20$) in seconds. Results are averaged across 4 runs on an Ubuntu 22.04 LTS desktop equipped with an AMD Ryzen 7 3700X 8-core CPU and 32GB DDR4 RAM.

| PE scheme | ZINC (↓) | CYCLES (↓) | PATTERN (↓) |
|---|---|---|---|
| LAPE | 13.79 | 28.59 | 886.46 |
| RW | 28.93 | 49.61 | **446.80** |
| GAPE | **8.34** | **19.40** | 769.21 |

**GAPE**$^{**}_{max}$ Like GAPE**, but use a different node label for each node ($m = n$) and initialize each initial weight vector to a different random vector.

The rationale behind ∗ versions of GAPE is that it is counter-intuitive for the transition weights of an automaton to be negative, and so normalizing the rows of $\mu$ should give a more probabilistic interpretation of the WGWA's traversal of the graph. The rationale behind ∗∗ is to normalize the columns of $\alpha$ according the PPR understanding of GAPE; teleportation probabilities cannot be negative.

Finally, recall that $\alpha$, the matrix of initial weights from Eq. (2), assigns different initial weights to each node label. While our base variation of GAPE uses only one node label, we also tried using $m = 20$ labels. The hope is that varying the initial weights can help GAPE learn on graphs with high levels of symmetry, such as those from CSL. By assigning each node a different initial weight vector, the automaton should be forced to compute different weights for nodes with similarly structured neighborhoods. We also experiment with giving every node a unique initial weight vector by letting $m = N$, where $N$ is larger than the size $n$ of any graph in the given dataset, and giving every node a different node label.

### 4.2.4 Discussion

The graph tasks reveal the shortcomings of the vanilla transformer. Without a PE, it is unable to adequately distinguish non-isomorphic graphs in CSL, and also struggles to detect 6-cycles and planar/non-planar graphs. All other PEs are able to improve on the baseline transformer, except on the CSL task where most PEs do not achieve 100% accuracy.

It is interesting that SA underperforms relative to LAPE on many graph tasks given how its approach positions it as an improvement on LAPE and performs better than LAPE when incorporated in the spectral attention graph transformer devised by Kreuzer et al. (2021). While the encoder used to create the PE may need more parameters, a more likely explanation is that performance suffers from a lack of incorporating edge embeddings, which the spectral attention graph transformer assumes but we omit for a fairer comparison of the node PEs alone.

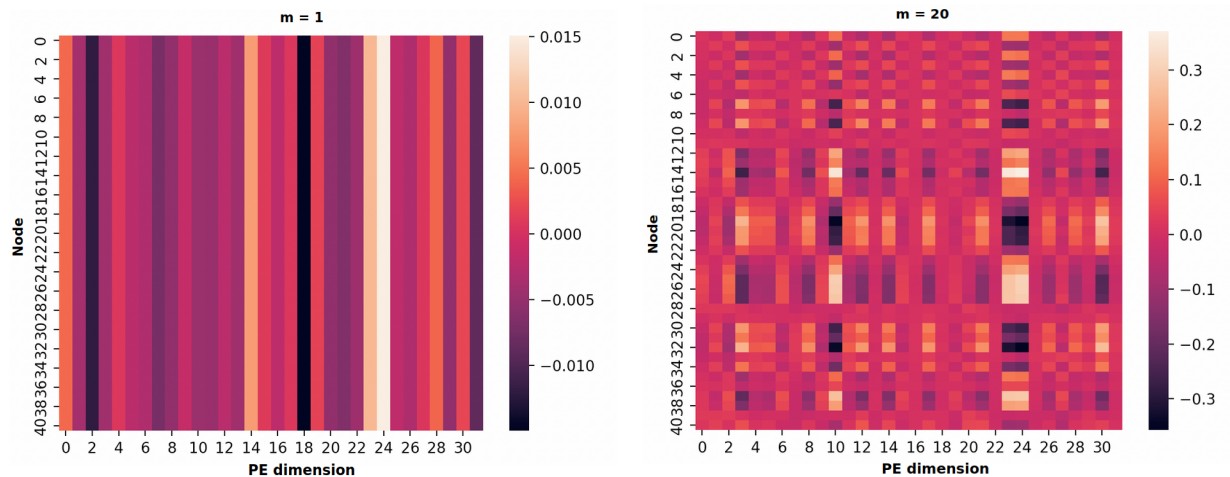

Figure 2: Comparing GAPE with varying number of node labels $m$. Left is GAPE with $m = 1$, and right is GAPE with $m = 20$. Nodes are ordered according to the order in which they were inserted into the graph when CSL was constructed.

We think it is natural to see that RW performs nearly perfectly on finding 6-cycles since a 20-hop neighborhood is enough to uniquely encode nodes in those 6-cycles, but performance begins to fall on CYCLES-V as expected as a 9-hop neighborhood is unable to provide a unique encoding to nodes in cycles of length greater than 9.

GAPE's ($\gamma = 0.02$) performance surpasses other PEs only on PATTERN and CLUSTER, with the other variants showing more competitive performance in other datasets. We give a discussion on the various factors causing these trends in the following subsections.

Table 3 compares the runtime for computing LAPE, RW, and GAPE according to three different sized datasets. ZINC, CYCLES, and PATTERN are increasing in average graph size. ZINC has 12K graphs with 23.16 nodes on average; CYCLES has 11.2K graphs with 48.96 nodes on average; PATTERN has 12K graphs with 118.89 nodes on average.

We see that GAPE is faster to compute for smaller graphs, but RW overtakes both LAPE and GAPE for large graphs. This is likely because RW is computable via matrix powers while an eigendecomposition for LAPE and use of the Bartels-Stewart algorithm for GAPE are more computationally demanding. Despite GAPE's theoretical time complexity being at least as large as that of LAPE, we suspect that GAPE's advantage regarding smaller graphs is due to efficiencies in the Sylvester equation solver.

### 4.2.5   Effect of number of states on performance

One thought for improving performance of GAPE on the graph tasks is to increase the number of states used in the WGWA on which it is based. However, we observed no positive trend in performance as we increase the number of states. For most tasks, increasing the number of states can even worsen performance. Table 4 shows performance on ZINC, CYCLES, CLUSTER, and PATTERN as the number of states $k$ varies using the base GAPE method in Table 2 with damping factor $\gamma = 0.02$. One thought is that adding states to the automaton adds risk of instability; if the GAPE encoding of a node $u$ is the total weight of all runs from any initial configuration to $(u, r)$ for all possible $r$, then adding states means adding more possible runs which may lead to weights that are too large, depending on the initialization of $\mu$.

### 4.2.6   Effect of number of initial weight vectors

Another thought is to consider how performance varies as the number of node labels $m$ increases. We see in Table 2 that increasing $m$ from 1 to 20 grants noticeable but not very large performance gains on nearly all

Table 4: GAPE Performance as $k$ varies

| $k$ | ZINC ($\downarrow$) | CYCLES ($\uparrow$) | CLUSTER ($\uparrow$) | PATTERN ($\uparrow$) |
|-----|------|--------|---------|---------|
| 8   | 0.281 | 79.70 | 72.41 | 85.76 |
| 16  | 0.279 | **86.80** | 72.55 | 78.24 |
| 32  | **0.251** | 79.03 | **73.40** | **85.99** |
| 64  | 0.278 | 80.36 | 69.90 | 75.52 |
| 128 | 0.269 | 82.28 | 72.14 | 79.49 |

datasets except for CSL. We think this is because increasing $m$ forces GAPE to assign different initial weights for runs starting at otherwise seemingly identical nodes. In other words, increasing $m$ is akin to conditioning the automaton's starting weights on distinct node labels on the graph. Such a labeling allows GAPE to distinguish between nodes with the same neighbors and nodes with neighbors with the same features, aiding in classifying highly symmetric graphs like CSL graphs. This explanation applies to the cycle detection datasets as well as increasing $m$ presents up to a 3.36% increase in accuracy on CYCLES-V.

We further note that when assigning every node a unique initial weight vector in a consistent fashion as in $\text{GAPE}^{**}_{\text{max}}$, we see further improved performance on many of these highly symmetric graph datasets, most notably achieving 100% on CSL. However, on none of these GAPE variants where $m > 1$ do we see a performance gain on PATTERN, CLUSTER, or PLANAR; we instead see a significant amount of overfitting. An explanation is that increasing $m$ may encourage the transformer to rely too heavily on each node's initial weight vectors to memorize node positions, leading to poor generalizability on graphs that do not have a highly symmetric structure. In other words, increasing $m$ strengthens the transformer's notion of node "identity" which is useful for distinguishing nodes in symmetric substructures like atom rings and cycles but less useful for tasks where nodes are part of a variety of less symmetric substructures.

Fig. 2 illustrates more clearly the effects of increasing $m$ for CSL graphs. We see that setting $m = 1$ results in assigning the same PE to every node in the CSL graph, preventing the transformer from learning the task. In contrast, setting $m = 20$ diversifies the PE, allowing the graph transformer to distinguish between identical neighbors and neighbors with identical features.

## 5  Limitations and Conclusion

While GAPE and its variants are competitive on certain datasets, they struggle to outperform the other PEs on datasets like CSL, CYCLES, and CYCLES-V without setting $m > 1$. Further, even with $m > 1$, GAPE's performance worsens on other datasets. Improvements on GAPE should seek to improve performance on data involving highly symmetrical structures like cycles, and more work needs to be done to ensure GAPE is able to retain performance even as $m$ increases. As it stands, it seems doubtful that a transformer equipped with GAPE with $m = 1$ should be more powerful than the 1-WL test, and there is no single variant of GAPE that results in better performance overall.

Another limitation of GAPE is the computational difficulty of solving for $P$. Using the "vec trick" allows us to learn $\mu$ and $\alpha$, but runs far too slow and consumes too much memory. We run out of memory when testing PATTERN with a batch size of 26 and $k = 32$, and shrinking the batch size to 18 and number of states to $k = 10$ results in a single epoch taking over an 1.25 hours, making learning $\mu$ and $\alpha$ impractical with this method. Keeping $\mu$ and $\alpha$ random means giving up possible performance gains and stability. We leave alleviating limitations to future work.

To sum up, GAPE provides a more theoretically principled framework for developing graph PEs. It is a generalization of the original sinusoidal encodings on strings, and it has connections with personalized PageRank and LAPE under certain modifications. With this, GAPE can be seen as a step towards bridging the spatial and spectral frameworks of graph PEs under graph automata.

## Acknowledgements

We thank the anonymous reviewers for their valuable comments. This material is based upon work supported by the US National Science Foundation under Grant Nos. IIS-2137396 and IIS-2146761.

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
