# OpenReview forum: "Bridging Graph Position Encodings for Transformers with Weighted Graph-Walking Automata"
_TMLR — Accepted by TMLR_

### Review · Reviewer_NPZV · 2023-01-16

**Summary Of Contributions:**

This paper proposed a new graph PE, Graph Automaton PE (GAPE), based on weighted graph-walking automata, which can generalize several other PEs.

**Audience:**

Yes

**Claims And Evidence:**

No

**Requested Changes:**

Refer to the "Weaknesses & Questions".

**Strengths And Weaknesses:**

### Strengths

- It seems the proposed positional encoding can generalize to the existing positional encodings.

### Weakness & questions
- The motivation is not clear.
  -  It is not clear why the author proposed "weighted graph-walking automata". The author claimed there are some shortcomings in previous Positional Encodings, but how the proposed "weighted graph-walking automata" addressed these shortcomings?
- The method section is not clear.
  - From Sec. 3.1, it is not easy to understand what are "graph-walking automata" and "weighted graph-walking automata". It would be better if some examples/figures are provided.
  - Is the "w" in eq(1) a vector or scalar? I am assuming it is a scalar.
  - It is unclear to me, how Sec. 3.2. connects to Sec. 3.1. Why do we need to solve the eq.(2)?
  - Is the $k$ in Sec. 3.1 the same as in Sec. 3.2?
- The experiment looks not impressive. From table 3, the proposed method can only outperform previous baselines in 2/8 tasks. Besides, some top 1 results are not marked in bold.
- The dataset used in experiments seems very small, for example, 12K subsample graphs from ZINC, 150 graphs from CSL. Previous works, like graphormer, usually use large-scale dataset, like PCQM4M. I think there should be some experiments on large-scale datasets to demonstrate the proposed method is better.

---

> ### Author Response · Authors · 2023-02-09
> **Thank you for your review**
>
> Thank you for the specific feedback. Please see our responses below to clarify our motivation, methods, and experiments.
> ## Motivation
>
>  We will add a clearer explanation of the motivation and benefits of GAPE. The main motivation behind GAPE is to encompass the other PEs with a method that is a generalization of them and is more principled. We hope that GAPE can give a more unified view of these other PEs for graphs and strings. We also address the shortcomings of spectral PEs in section 1 by showing that GAPE is able to operate on directed graphs, and further provides a distributed representation for a PE unlike RW and LAPE.
>
> ## Methods
>
> We will add a clearer explanation of the connection between equations 1 and 2. The idea is this: suppose we have runs of the form $(q_1, v_1), \dots, (q_T, v_T)$ as defined in Definition 3. Let $\mathcal{R}_{r, v}$ be the set of all such runs of a WGWA $M$ on a graph $G$ ending in configuration $(r,v)$.
> Define the **forward weights** $P \in \mathbb{R}^{k \times n}$ as
>
> $$
> P_{r, v} = \sum_{(q_1,v_1), \ldots, (q_T, v_T) \in  \mathcal{R}_{r, v}}  (\alpha \ell)\_{q_1, v_1} \\ \mu_\{q_1, q_2}  \cdots   \mu_\{q_\{T-1}, q_\{T}}
> $$
>
> $$
> = \sum_{(q_1,v_1), \ldots, (q_T, v_T) \in \mathcal{R}_{r, v}} (\alpha \ell)\_{q_1, v_1} \prod_\{t=1\}^\{T-1} \mu_\{q_t, q_\{t+1}}
> $$
>
> which is the total weight of all run **prefixes** ending in configuration $(r,v)$.
> We can rewrite this as
>
> $$
> P_{r,v} = (\alpha \ell)\_{r,v} + \sum_{\substack{(q_1,v_1), \ldots, (q_T, v_T) \in \mathcal{R}_{r, v},\\ T > 1}} (\alpha \ell)\_{q_1, v_1} \prod_\{t=1\}^\{T-1} \mu_\{q_t, q_\{t+1}}
> $$
>
> $$
> = (\alpha\ell)\_{r,v} + \sum_{q,u}\left\(\sum_{(q_1,v_1), \ldots, (q_{T-1}, v_{T-1}) \in \mathcal{R}_{q, u}}(\alpha\ell)_\{q_1, v_1} \prod_\{t=1}^\{T-2} \mu_\{q_t, q_\{t+1}}\right) \mu_\{q,r} A_\{u,v}
> $$
>
> $$
> = (\alpha\ell)\_{r,v} + \sum_{q, u} \\ P_{q, u} \\ \mu_{q, r} A_{u, v}.
> $$
>
> The matrix version of the above $P = \mu P A + \alpha \ell$. This and the above assumes all entries of $\tau = 1$, and so the matrix version of the above becomes $P = \mu PA + \alpha\ell$ as seen in equation 2. We will include all of this in Section 3.2.
>
> $w$ in eq(1) is indeed a scalar, and the $k$ in Sec. 3.1 is the same as the $k$ in Sec. 3.2.
>
> ## Experiments
>
> While GAPE does not outperform the majority of the previous PEs, we believe that the value of GAPE lies more in its theoretical properties as a generalization of other PEs and as a more principled approach. Regarding experiments on larger datasets, we are currently finishing experiments with OGB's PCQM4Mv2, the results of which we will include.

---

> > ### Comment · Reviewer_NPZV · 2023-02-23
> > **thank you for the response!**
> >
> > - for motivation. I don't understand why "RW and LAPE" cannot provide a distributed representation. Using an integer to index a distributed representation from a matrix (so-called embedding lookup) is widely used in deep learning.
> > - I saw the paper update the PCQM4Mv2 results, but it seems the performance is inferior, and has a large gap compared with graphormer (~0.087 in the validation set). I am not sure why you report OOM for "SPD+C". Besides, there are some numbers like "0.914" that looks like the model is failed.

---

> > > ### Author Response · Authors · 2023-02-24
> > > **Thank you for the feedback**
> > >
> > > Thank you for reviewing the updated paper and the additional feedback.
> > >
> > > * RW and LAPE and distributed representations: we would like to point out that neither RW nor LAPE actually index an embedding lookup as is visible in Section 3.3.2 and Section 3.3.3. They are non-distributed features simply added to the node feature matrix.
> > > * Graphormer and why we report OOM: we did not report Graphormer's validation MAE because SPD+C does not refer to Graphormer. SPD+C refers only to the spatial and centrality encodings of Graphormer which we use in our transformer. Graphormer uses additional methods to improve performance, namely edge encodings and a virtual node for graph pooling which is connected with every other node in the graph. We omit these methods in order to experiment with only the node PEs for a fair comparison with the other node PEs. We report OOM for SPD+C because we ran out of memory during training; we used a combination of NVIDIA 1080 Ti (11GB VRAM) and TITAN X (12GB VRAM) GPUs.
> > > * Performance on PCQM4Mv2 and high MAE metrics: while variants of GAPE certainly underperform, the point is to show the effects of various changes to GAPE which we detail in Section 4.2.3.

---

### Review · Reviewer_Pqa9 · 2023-01-19

**Summary Of Contributions:**

This work presents a method for computing node embeddings for graph neural networks, motivated by the need for positional embeddings in transformers. As far as I understand it, the method is based on summing over the weights of all paths of of an automaton that (a) start at any node with a node-label-dependent initial state, (b) repeatedly transition to any neighboring node, (c) simultaneously (but independently of the node or edge traversed) transition to a different state, (d) end at any node with a node-label-dependent final state. Mathematically, this involves solving a linear fixed-point equation with a solver algorithm, written in terms of Kronecker products.

The authors argue that this approach generalizes a wide number of existing positional encodings, but I believe there are correctness issues with many of their claims. *(Note: Most of these issues have been resolved in the updated submission, see comments below.)*

The authors show that their method can learn to reproduce a fixed embedding on a fixed graph, and that it learns well-performing embeddings for sequences if sequences are interpreted as directed graphs. They also present results on a number of synthetic datasets of graphs, finding that it performs fairly similarly to previous methods, and explore how variants of their method affect performance.

**Audience:**

Yes

**Broader Impact Concerns:**

No broader impact concerns.

**Claims And Evidence:**

Yes

**Requested Changes:**

## Important requested changes
This paper has a number of issues that should be addressed before I would recommend acceptance for this work.

### Correctness issues
As I stated above, the paper has a number of correctness issues regarding its connection to LAPE and PPR. I would like to see those issues fixed.

### Recovery of LAPE embeddings
I think section 4.1 is somewhat misleading due to the presence of trivial solutions for any embedding of interest on a fixed graph, not just LAPE. If the authors want to argue that their approach provides the same benefits as LAPE, they need to show that it simulates LAPE on new graphs that it was not trained on. Otherwise, I think keeping this section would just be confusing to the reader, and I would suggest removing it.

### Statements about incorrectness of prior work
The authors state that they "show to be false" the claims in Dwivedi et al. (2020) about Laplacian eigenvectors being a generalization of Transformer sinusoidal embeddings. The evidence for this appears to be that models trained using Laplacian-based positional encodings do not perform well at machine translation, whereas sinusoidal position encodings do perform well. In my view this is a strong statement that isn't supported by a lot of evidence, and is unnecessarily adversarial toward prior work.

Looking at Dwivedi et al. (2020), the claim there is that the eigenvectors of the line graph are cosine and sine functions, and that this is a "natural generalization" of the Transformer's embeddings. The authors here also state that a cycle graph's eigenvectors are also sinusoidal. So, what is the claim being disputed here? Whether they are sinusoidal or not? Whether being sinusoidal is enough to count as a generalization? Whether they produce the same performance in a specific case?

(If the authors just want to state that, despite being sinusoidal as Dwivedi et al. (2020) say, LAPE/SA encodings don't perform as well as a different type of sinusoidal embeddings, I think it would be clearer to focus on that specific claim instead of on the accuracy of prior work.)

## Other suggestions and comments

- Some additional related work that I think would be worth discussing:
  - [Learning Graph Structure With A Finite-State Automaton Layer (Johnson et al. 2020)](https://arxiv.org/abs/2007.04929), which also uses automata for processing graphs and uses a linear system to compute embeddings,
  - [The Graph Neural Network Model (Scarselli et al. 2009)](https://ro.uow.edu.au/cgi/viewcontent.cgi?article=10501&context=infopapers), an early approach to graph neural networks which also involved solving for a fixed point (but interpreted it as a fixed point of message passing rather than a sum over automaton weights), and which seems closely related mathematically.
  - [Convergent graph solvers (Park et al. 2022)](https://arxiv.org/abs/2106.01680), another more recent approach to graph neural networks using fixed points
- Section 3.1 mentions that the approach uses directed graphs but also that the graphs do not have "directions", this is a bit confusing.
- It was difficult to follow why (2) gives "the weights for all possible r"; it might be worth explaining this more intuitively.
- I'm not sure what the authors mean by "we pass $P^T$ through a linear layer"?
- I don't understand what the runtime of the algorithm has to do with the orthogonality of $\alpha$ at initialization.
- In 3.3.2, statements about "the eigenvectors" and "the solution" should be more precise, since there are multiple solutions and eigenvectors can appear in multiple orders.
- In 3.3.3., $\beta$ is described as giving the probability of terminating the walk, but I believe it actually gives the probability of teleporting back to the start node? (Teleportation is later referred to in section 4.3.3 without being introduced beforehand.)
- It wasn't clear to me why the PPRP encoding is interesting, and the method for computing it (by taking diagonals of a bunch of powers) seems just as complex as computing RW itself, without GAPE adding any value here.


**Strengths And Weaknesses:**

$
\newcommand{\R}{\mathbb{R}}
$

## Strengths
- The idea of using an automaton to construct positional embeddings is an interesting one, and there do seem to be some interesting relationships between their proposed embeddings and some that have been proposed previously.
- The authors conduct some informative ablations of their technique, showing how its expressivity varies based on the number of states and on the impact of introducing random labels.

## Weaknesses
- The theoretical claims seem to have a number of correctness issues. The authors argue that their approach is a generalization of Laplacian eigenvector encodings, but I believe their derivation is incorrect. I also believe there are (fairly minor) issues with their connection to Personalized Page Rank.
- The proposed approach is based on an automaton that walks the graph and switches states. However, the state transitions are independent of the node's position on the graph or its movement, and only seem to influence the number of steps that the automaton takes before stopping. This seems like a big limitation for the expressivity of this approach, since the automaton is essentially doing a "blind" random walk. (Note that the graph-walking automata defined by Kunc & Okhotin (2013), the stated starting point for the approach proposed here, gain their expressive power by using node and edge labels to influence transitions.)
- Section 3.2, which defines the method, is quite difficult to follow. What P means, or why it satisfies (2), aren't clearly explained or well motivated.
- The proposed algorithm appears to require a fairly-expensive matrix-inversion step, which might make it impractical to use as a position embedding for large graphs. I'm also not sure that scaling $\mu$ by a hyperparameter at initialization is sufficient to ensure that the method doesn't break later in training due to bad conditioning.
- Their empirical validation of their method's ability to recover Laplacian eigenvector encodings seems insufficient. They show that their method can replicate the other encodings for a fixed graph, but there is a trivial solution for recovering any encoding for a fixed graph, so this doesn't mean much.
- Some details regarding the way their approach is actually used are unclear. For instance, their method takes as input some vectors of "initial weights" which are indexed by the nodes in the graph. If this is to be used as a positional encoding for new graphs, how are these initial weights determined? Are they learned for each graph separately, or are they produced by a network?
- Some related work is not cited.
- Their experimental results are not particularly strong and are primarily on synthetic data.

## Additional comments on weaknesses

### Correctness of the connection with Laplacian eigenvector encodings (3.3.2)
The authors claim their approach generalizes LAPE, by arguing that the output of LAPE is also the output of their approach given a particular configuration of inputs. However, I do not think this is actually true.

Specifically, the way the authors compute GAPE, described in section 2, is to solve a linear system, given the result $\alpha \ell$ of a matrix product. In 3.3.2, the authors rewrite the LAPE embedding to resemble the form of the GAPE embedding. However, this rewriting has $\alpha \ell = 0$! This means that there is not a unique solution to the (modified) GAPE equation, and the linear solver will not necessarily produce the one corresponding to LAPE. (Indeed, it is likely to produce a zero vector instead.) Actually recovering the LAPE output would require running an eigenvector computation algorithm instead of solving a linear system.

Even ignoring this issue, there are also other problems that the authors discuss. Specifically, the transition matrix now depends on the graph Laplacian, and so would not generalize to new graphs, and the modified equation now uses the Laplacian instead of an adjacency matrix, making it not correspond well to an automaton.

### Correctness of connection to Personalized Page Rank
The authors similarly attempt to show that their method is equivalent to PPR. However, I don't think this claim is accurate either. Specifically, although equations (2) and (4) are structurally similar, the matrices are the wrong dimensions. The authors set $k = 1$ and then write $\alpha = \beta I$, but $\alpha \in \R^{k \times m} = \R^{1 \times m}$ and $\beta I \in \R^{m \times m}$. Additionally, the term for $\ell$ appears to be missing.

Perhaps the authors meant to specify $\alpha \ell = \beta I$, with $k = m = n$, and $\ell = \mu = I_{n \times n}$? This would correspond to having each node getting a different label, every label leading to a different starting automaton state, and the automaton always staying in the same state?

### Ability to recover LAPE in practice
Although the authors present some empirical results on the ability of their method to recover LAPE, I didn't find their methodology convincing, and I don't think their actual findings here mean much. Specifically, they test whether, given a specific graph and its LAPE embeddings, they can learn to approximate those specific embeddings using GAPE. However, as the authors note, there is a trivial solution to this for ANY embedding (not just LAPE): set $\mu = 0$ and $\alpha$ as the desired embedding.

The authors argue that their approach doesn't learn this particular trivial embedding, but there are likely many other equally-trivial solutions. In order to be convincing, this experiment would have to verify that GAPE can match the LAPE embeddings for new unseen graphs, without having a per-graph learning step.

---

> ### Author Response · Authors · 2023-02-09
> **Thank you for your review**
>
> Thank you for the very thoughtful review, especially regarding the correctness of the LAPE encoding. Please find our responses and clarifications below.
>
> ## Correctness of the connection with LAPE and its recovery
>
> Your thorough analysis is well-received, and we are also convinced that the results in learning LAPE for fixed graphs are too weak to suggest GAPE is learning a non-trivial solution besides the one where $\mu = 0$, let alone for unseen graphs. Section 4.1 will be removed.
>
> ## Correctness of the connection with Personalized PageRank
> Thank you for pointing out the issues with the matrix dimensions; you are correct, $\alpha\ell = \beta I$ with $k = m= n$ and $\ell = \mu = I$. Your interpretation of the corresponding is also correct. We also need to mention that $A$ in Equation 2 needs to be substituted with the 1-hop random walk matrix $W$ which corresponds to the automaton walking the graph by starting on a vertex and repeatedly selecting one of its neighbors to visit with equal probability.
>
> ## Statements about incorrectness of prior work
> We will amend our wording and clarify the relationship between this work and and the claims made by Dwivedi et al. (2020); being adversarial was certainly not our intent. Our main point is what you describe, namely that despite being sinusoidal as Dwivedi et al. (2020) say, LAPE/SA do not recover the performance of the sinusoidal encoding.
>
> An additional point is that, regarding the ability to generalize the original sinusoidal encodings, GAPE stands out with empirical and theoretical advantages. It nearly recovers the performance of the sinusoidal encodings in MT, and there is a setting of initial, transition, and final weights which recovers the sinusoidal encodings.
>
> We also realize that only testing LAPE of the cycle graph may be confusing, so we conducted an additional experiment on MT with LAPE using the path graph of $n$ vertices and obtained a BLEU of 17.3. We will include this in the paper, and we will reword and clarify our claims.
>
>
> ## Other comments
> - How initial weights are determined: In our experiments, the initial weights are randomly generated independent of the input graph. However, one could make them learnable, provided they have a differentiable method to solve for $P$, or they could use a separate network as you describe.
> - Related work: Thank you for bringing these works to our attention; we will include discussion of them.
> - Directions: "Directions" in Kunc \& Okhotin (2013) refer to labels of the endpoints of edges; i.e., they indicate the ends of arrows in a graph. For clarity, since they are not important for GAPE and we do not refer to these anywhere else in the paper, we have removed their use.
> - Equation 2 and "for all possible $r$": In our response to reviewer NPZV, we give a clearer connection between equations 1 and 2 which hopefully clears up the confusion for how equation 2 gives the weights of all possible $r$. We also hope it better motivates and explains why equation 2 follows from equation 1.
> - Linear layer: If $d$ is the number of node features, and $k \neq d$, then we pass $P$ through a linear layer so that its dimensions match with the node feature matrix.
> - Runtime and initialization: As you mention, one way to solve for $P$ is through a matrix inversion step. However, the Bartels-Stewart algorithm is able to solve for $P$ more efficiently, but no implementation in PyTorch is available, so $\alpha$ and $\mu$ are not learnable and must remain random. Our choice to initialize $\mu$ and $\alpha$ using orthogonal initialization is inspired by Saxe et al. (2014) who show that orthogonal inputs are better suited for deep networks.
> - Eigenvectors and wording: We will make the statements about "the eigenvectors" and "the solution" more precise.
> - $\beta$ as teleporting probability: Thank you, we will make this correction.
> - GAPE and PPRP: The point is that the method used to produce the RW encoding can also be used on a special case of GAPE to produce a similar encoding which looks like and performs similar to RW. In this way, GAPE can be seen as a kind of generalization of RW.

---

> > ### Comment · Reviewer_Pqa9 · 2023-02-10
> > **Additional clarification questions**
> >
> > Thanks for the response. A few more questions based on your comment:
> >
> > > we pass $P$ through a linear layer so that its dimensions match with the node feature matrix.
> >
> > I'm still not sure what you mean by this. $P$ is an intermediate value (of shape $k \times n$) inside the computation of $GAPE$, is it not? Why does it need to match the node feature matrix?
> >
> > One guess: perhaps $GAPE_M(v)$ is what is passed through a linear layer, as an implementation detail of the transformer model? If so, it's confusing that the linear layer refers to $P$ itself, since $P$ is an input to the equation defining $GAPE$.
> >
> > > the Bartels-Stewart algorithm is able to solve for $P$ more efficiently, but no implementation in PyTorch is available, so $\mu$ and $\alpha$ are not learnable and must remain random.
> >
> > Why does the availability of a PyTorch implementation influence whether $\mu$ and $\alpha$ are learnable?
> >
> > Ah, do you perhaps mean that you still use Bartels-Stewart but in a non-differentiable way, so gradients are not available for $\mu$ and $\alpha$? If so, I think that sentence in the paper could be reworded to clarify that it's the non-differentiability of the specific solver that you are using that prevents learning of $\mu$ and $\alpha$. (It's also not stated in the paper that $\mu$ and $\alpha$ aren't learned; you just mention how you initialize them. When I read the paper I had assumed they were learned, and that you used a different solver that was available in PyTorch.)
> >
> > On a related note, have you explored either differentiating through the matrix inversion version in PyTorch, or using an implicit differentiation strategy similar to https://jax.readthedocs.io/en/latest/_autosummary/jax.lax.custom_linear_solve.html ? (I'm not sure what the equivalent is in PyTorch.)
> >
> > > GAPE and PPRP: The point is that the method used to produce the RW encoding can also be used on a special case of GAPE to produce a similar encoding which looks like and performs similar to RW. In this way, GAPE can be seen as a kind of generalization of RW.
> >
> > Personally I don't find this argument very convincing, since the only part of RW that is being substituted is the matrix $W = AD^{-1}$, and most of the complexity of the RW encoding seems to come from the postprocessing performed afterward. (As an extreme example, I wouldn't call the matrix $W$ itself a generalization of RW either.)

---

> > > ### Author Response · Authors · 2023-02-14
> > > **Responses to additional questions**
> > >
> > > Thank you for the comments and questions.
> > > > One guess: perhaps $GAPE_M(v)$ is what is passed through a linear layer, as an implementation detail of the transformer model?
> > >
> > > Yes, this is the idea. We will clarify this in Section 3.2.
> > > > Ah, do you perhaps mean that you still use Bartels-Stewart but in a non-differentiable way, so gradients are not available for $\mu$ and $\alpha$?
> > >
> > > Yes, this is what we mean. We will reword this section in Section 3.2 to clarify the relationship between the Bartels-Stewart solver and the non-learnability of $\mu$ and $\alpha$.
> > > > On a related note, have you explored either differentiating through the matrix inversion version in PyTorch, or using an implicit differentiation strategy similar to https://jax.readthedocs.io/en/latest/_autosummary/jax.lax.custom_linear_solve.html?
> > >
> > > We have experimented with differentiating through the matrix inversion, but we found the runtime to be intractably long. We mention some runtime issues in Section 5; it just took too long to train even when shrinking batch sizes and $k$. Thank you for bringing up implicit differentiation; we were not aware of any PyTorch implementations. After some research, it seems [TochOpt](https://pypi.org/project/torchopt/) has a way to compute implicit gradients. It would be interesting to explore GAPE with learned $\mu$ and $\alpha$ with this method.
> > >
> > > > Personally I don't find this argument very convincing, since the only part of RW that is being substituted is the matrix $W = AD^{-1}$, and most of the complexity of the RW encoding seems to come from the postprocessing performed afterward.
> > >
> > > Saying GAPE is a generalization of RW may then be too strong; a reduced claim which still adds value would be that GAPE, through it's connection with PPR and therefore PPRP, can be made analogous and empirically equivalent to RW through similar postprocessing. GAPE's connection with RW is admittedly weaker than its connection with PPR. We will reorganize and reword Section 3.3.3 to better clarify GAPE's value in its connections with PPR and RW.

---

> ### Comment · Reviewer_Pqa9 · 2023-02-19
> **Response to updated paper**
>
> I have read the new version of the paper, which resolves many of my concerns. A few relatively minor additional comments:
>
> - In my opinion, it's still not quite right to say that GAPE "is a generalization of other graph PEs including personalized PageRank (Page et al., 1998) and the Laplacian encoding by Dwivedi et al. (2020) under certain assumptions". In particular, actually recovering PPR or the LAPE encoding requires modifying the formula for GAPE (and, for LAPE, also modifying the algorithm used to solve the formula), so the version of GAPE presented in sections 3.1 and 3.2 doesn't generalize those other PEs. This is more than an "assumption", it's a change in the method.
>     - The new version of the paper is much clearer about what changes need to be made to get from GAPE to LAPE/PPR, so I don't think this is a major flaw, but I still think it's a bit of an inaccurate claim.
> - Thanks for providing the derivation of P in section 3.2. One thing that might be a mistake: should there be a transpose for $\mu$? It looks like the equation above is contracting across the first index of $\mu$ ($\sum\_{q,u} P\_{q,u} \mu\_{q,r} A\_{u,v} = \sum\_{q,u} (\mu^T)\_{r,q} P\_{q,u} A\_{u,v}$), but equation (2) is written with a non-transposed $\mu$. (Alternatively, perhaps $\mu$ should be interpreted backwards, and written as $\mu_{r,q}$ in the derivation?)
> - The wording of the "negative case" of the PLANAR dataset description is a bit odd to me. Although some non-planar graphs can be detected by checking whether $|E| > 3|V| - 6$, some non-planar graphs cannot be detected this way and require the same homeomorphism check as the "positive case". This would probably be clearer if you said something like "Many non-planar graphs can be detected merely by counting nodes ... . However, exactly checking whether a graph is non-planar requires checking whether there are subgraphs homeomorhpic ..."
> - Figure 2's caption could use a bit more detail and the figure would benefit from axis labels. I assume that different rows are embeddings of different nodes? Are the nodes ordered in some way?

---

> > ### Author Response · Authors · 2023-02-23
> > **Response to additional comments**
> >
> > Thank you for looking over the updated paper and the additional feedback.
> >
> > > The new version of the paper is much clearer about what changes need to be made to get from GAPE to LAPE/PPR, so I don't think this is a major flaw, but I still think it's a bit of an inaccurate claim.
> >
> > One of the main modifications to GAPE needed to recover LAPE/PPR is replacing the adjacency matrix $A$ with a different $n \times n$ matrix. What may help address your concern is that the choice of such a matrix results in conceptually minor changes to GAPE. Substituting $A$ for a weighted adjacency matrix (like $W$ for example) in Equation 2 only changes how the automaton walks the graph, but the core method for computing a PE is unchanged. Rewording GAPE to use any weighted adjacency matrix may bring it closer to being a full generalization of other PEs.
> > Admittedly, as we note in the paper, interpreting the Laplacian in terms of GAPE is left unclear.
> >
> > > Should there be a transpose for $\mu$?
> >
> > Thank you for the correction; $\mu$ should be transposed.
> >
> > > The wording of the "negative case" of the PLANAR dataset description is a bit odd to me.
> >
> > Thank you for the suggested wording. We agree it is clearer and more accurate. We will update our description of PLANAR.
> >
> > > I assume that different rows are embeddings of different nodes? Are the nodes ordered in some way?
> >
> > The rows are indeed embeddings of different nodes, which we indicate in Figure 2's caption. There is no special ordering of the nodes. They are ordered according to the order in which they were inserted into the graph when the CSL dataset was constructed. We will clarify this and label the axes of all heatmaps in the paper.

---

### Review · Reviewer_kBUH · 2023-01-24

**Summary Of Contributions:**

The success of transformers in language, vision, etc. has given rise to interest in using them for graph-structured data. However, the notion of what constitutes position and positional encoding in this setting is unclear. This paper proposes to use the formulation of weighted graph-walking automata (WGWA) to derive positional encodings (PEs) called graph automaton positional encoding (GAPE). They show how this connects to sinusoidal encodings, LAPE, personalized-pagerank and random walk based PEs. They experimentally compare GAPE with these and other PEs on machine translation and graph/node tasks.

**Audience:**

Yes

**Broader Impact Concerns:**

Nothing specific.

**Claims And Evidence:**

No

**Requested Changes:**

* Section 3.1: The authors say _We consider directed graphs_ and _Our graphs do not have initial nodes or “directions.”_. This is confusing because "directions" in directed edges are arrow heads. I am assuming that "directions" wrt initial nodes refers to the incoming edges to initial nodes that do not have source nodes. Unless it is essential, you can avoid using the term "directions" for the latter. Else please clarify the terms in the paper.
* The matrix $\ell$ can have only one 1 in each column. Please state this because $\ell$ as a matrix does not impose this constraint.
* What is $S$ in $q \in S$ in Def. 3?
* I believe the terms $\mu(\ell(u))_{qr}$ and $\tau(\ell(u))_{q,\ell q}$ have typos, in their use of $\ell(u)$ and $\ell q$ respectively.
* There other popular forms of PEs on linear data like strings, e.g., relative and rotary PEs. Any insights on how do they relate to GAPE?
* In the proof of Prop. 2, if $m = 1$ (unary) then why do $\alpha$ and $\tau$ have 2 columns?
* Why is the English to Vietnamese dataset selected for machine translation study?
* Last para, section 4.3.4: "GAPE's performance surpasses other PEs on PATTERN and CLUSTER." As per Table 3, on PATTERN, RW is the best.

**Strengths And Weaknesses:**

Strengths
* I found the principled approach of starting with a graph automaton and the reachability relation over its configurations using weighted transitions and labeling quite interesting.
* The connections of GAPE to some of the existing PEs can help improve our understanding of the space of PEs for graphs.
* The paper conducts experiments to compare GAPE with other PEs.

Weaknesses
* The actual linkage of GAPE with existing PEs is not strong. The result on sinusoidal encodings follows from DeBenedetto and Chiang (2020). PPR is also shown to be a special case. The connection to LAPE and PPRP/RW is demonstrated empirically.
* The experimental results show that GAPE can only outperform the baselines methods in 1/8 datasets. That dataset is machine-translation which is not really on graph-structured data.
* The authors start with simplest automata (single-state or unary alphabets) and extend the scope to some variants and multiple states and alphabet symbols. However, these explorations do not yield much benefits. Therefore, GAPE dos not show empirical benefits.
* GAPE can be computed in cubic time in the number of graph nodes or automaton states (possibly smaller). The paper doesn't report computing costs vs. other PEs.

---

> ### Author Response · Authors · 2023-02-09
> **Thank you for your review**
>
> Thank you for the helpful feedback. We address some of the weaknesses and requested changes below.
>
> ## Weaknesses
> - Experimental results and empirical benefits: While GAPE may not outperform other PEs in most datasets, we think its main contribution is in its theoretical properties as bridging other PEs under the lens of graph automata. We will also point out that GAPE also outperforms the baselines in CLUSTER.
> - Computing cost: We experimented with LAPE, RW, and GAPE by measuring their averaged computation time on an AMD Ryzen 7 3700x 8-core CPU over 4 runs. Our results are in the table below. We set $k = 20$, and experiment with three different sized datasets. ZINC has 12K graphs with 23.16 nodes on average. CYCLES has 11.2K graphs with 48.96 nodes on average. PATTERN has 12K graphs with 118.89 nodes on average.
> |  PE Scheme  | ZINC  |  CYCLES | PATTERN |
> | ----- | ----- | ----- | ----- |
> | LAPE  | 13.79  |  28.59  | 886.46 |
> | RW   | 28.93  | 49.61  | **446.80** |
> | GAPE | **8.34** | **19.40** | 796.38 |
> While this is just one aspect of computation to analyze regarding PEs, we see that GAPE is faster for smaller graphs, but RW overtakes both LAPE and GAPE for large graphs. This is likely because RW is computable via matrix powers while an eigendecomposition (LAPE) and use of the Bartels-Stewart algorithm (GAPE) are more computationally demanding. Despite GAPE's theoretical time complexity being at least as large as that of LAPE, we suspect that GAPE's advantage regarding smaller graphs is due to efficiencies in the Sylvester equation solver.
>
> ## Requested changes
>
> - Node directions: Thank you, for clarity we will remove using the term "directions."
> - Constraint on $\ell$: Thank you, we will mention this constraint.
> - Meaning of $S$: $S \subseteq Q$ is a set of initial states. We will update the definition of the WGWA to include this.
> - Typos with $\ell$: $\mu(\ell(u))\_\{q,r}$ is actually not a typo, but we need to add that $\mu$ is a map from $\Sigma$ to $k \times k$ matrices. For a unary alphabet, we just write $\mu$. The usage of $\ell q$ and $\ell(u)$ in $\tau(\ell(u))\_\{q,\ell q}$, however, is a typo. It should be $\tau_{q,\ell(u)}$. Thank you for bringing this to our attention.
>  - Rotary PEs: The rotary PE used in RoFormer by [Su et al. (2021)](https://arxiv.org/pdf/2104.09864.pdf) can actually be written in a similar form as our generalization of the original sinusoidal PE by [Vaswani et al. (2017)](https://arxiv.org/pdf/1706.03762.pdf). The difference is we set $\alpha = I$ and then replace the block matrices of $\mu$ with counter-clockwise rotation matrices like so:
>             $$\mu = \begin{bmatrix}
>             \cos \theta_1 & -\sin \theta_1 & 0 & 0 & \cdots \newline
>             \sin \theta_1 & \cos \theta_1 & 0 & 0 & \cdots \newline
>             0 & 0 & \cos \theta_2 & -\sin \theta_2 & \cdots \newline
>             0 & 0 & \sin \theta_2 & \cos \theta_2 & \cdots \newline
>             \vdots & \vdots & \vdots & \vdots & \ddots
>             \end{bmatrix}$$
>         where $\theta_j = 10000^{-2(j-1)/k}$. Then the PE for node $i$ is $\alpha\mu^i$ which can then be right-multiplied by the learnable key/query matrix and the $i$th embedding vector.
>
>  - Columns of $\alpha$ and $\tau$: In the beginning of Section 3.3.1, we define the graph of a string with 2 labels: one for identifying the first node and the other for the rest of the nodes. To clarify, we are only making a comparison with DeBenedetto \& Chiang (2020), not directly taking their approach with a unary alphabet.
>  - Why English to Vietnamese: This was the default dataset included in the used transformer implementation.
>  - Performance on PATTERN and CLUSTER: Two entries in the column are bolded since the difference between RW and GAPE's performances are not statistically significant. This is mentioned directly below Table 3. We used a t-test (p=0.05) to determine significance, which we will specify in the paper.

---

> > ### Comment · Reviewer_kBUH · 2023-02-18
> > **Thank you for your response**
> >
> > Thank you for your response and changes to the paper. I have looked through them.

---

### Decision · Action_Editors · 2023-03-12

**Recommendation:** Accept with minor revision

**Comment:**

Two reviewers lean towards rejection, while the most thorough reviewer leans towards acceptance. One of the main reasons to argue for rejection was unconvincing empirical improvements, but I found the writing to be modest enough that this doesn't lead to a strong mismatch between claims and evidence. I don't think this is grounds for rejection by TMLR standards.

The main reasons for rejection are the following (independently from two different reviewers):
* the claim that it "bridges" PE for graph transformers is not demonstrated well
* the framing of their method as a generalization of other PEs is a bit exaggerated

Given that the authors were receptive to addressing other issues that arose during review and that there seems to be consensus amongst authors and Reviewer Pqa9 about the technical details around how GAPE needs to be modified to recover LAPE/PPR, I believe we should extend some trust that the writing can be updated a bit more in the camera ready version to mitigate these concerns. Please see, for example, the last discussion with Pqa9 around, "I still think it's a bit of an inaccurate claim." I'd strongly encourage the authors to update the text in the camera-ready version to precisely describe the relationship in a way that the reviewers would describe as accurate.

**Audience:**

Yes. Positional embeddings for graph representations in Transformers is an important area, and there are some interesting new ideas for the area in the submission.

**Claims And Evidence:**

There were a number of issues with correctness and positioning in the initial submission, but Reviewer Pqa9 provided a very thorough review, and the authors were receptive to the feedback. After satisfying Reviewer Pqa9, I believe that the paper can be accepted by TMLR standards.

Other reviewers note that the empirical results are not particularly strong and the contribution is modest, but the paper is modest in the claims that it makes about the results, so this seems appropriate for acceptance.

Having said that, there is a remaining concern that the framing of their method as a generalization of other PEs is a bit exaggerated. Especially given the importance that TMLR places on claims being supported by accurate and clear evidence, I  strongly encourage the authors to tone down these claims in the camera-ready version.

---

> ### Author Response · Authors · 2023-03-28
> **Thank you**
>
> We greatly appreciate the valuable feedback from the reviewers and action editors. Below are the changes for the camera-ready version.
>
> - Reduced the claim that GAPE is a generalization of LAPE and PPR in sections 1 and 5
> - Further clarified the relationship between GAPE and LAPE & PPR in sections 3.3.2 & 3.3.3
> - Minor word choice changes
>
> The camera-ready version has been uploaded along with code.